Gene co-expression network for analysis of plasma exosomal miRNAs in the elderly as markers of aging and cognitive decline

http://orcid.org/0000-0003-2288-951X Ye Zheng
Sun Bo
Mi Xue
http://orcid.org/0000-0003-2288-951X Xiao Zhongdang zdxiao@seu.edu.cn
State Key Laboratory of Bioelectronics, School of Biological Science and Medical Engineering, Southeast University , Nanjing, Jiangsu , China
de Azevedo Walter Jr
Electronic publication date: 2020 Jan 6
Publication date: 2020
Volume: 8
Electronic Location ID: e8318
Received 2019 Aug 28; Accepted 2019 Nov 29
Copyright: © 2020 Ye et al.
Copyright year: 2020
Copyright holder: Ye et al.
License: This is an open access article distributed under the terms of the Creative Commons Attribution License, which permits unrestricted use, distribution, reproduction and adaptation in any medium and for any purpose provided that it is properly attributed. For attribution, the original author(s), title, publication source (PeerJ) and either DOI or URL of the article must be cited.
License URL: https://creativecommons.org/licenses/by/4.0/

Keywords: WGCNA, Plasma, Exsome, miRNA, Eldly, Aging, Cognitive decline, MoCA score, Biomarker

Funding: National Natural Science Foundation of China 81671807 Central Universities 2242018K3DN05 and 2242017K40236 This work was supported by grants from the National Natural Science Foundation of China (No. 81671807) and Fundamental Research Funds for the Central Universities (2242018K3DN05, 2242017K40236). The funders had no role in study design, data collection and analysis, decision to publish, or preparation of the manuscript.

==============================
Background

Evidence has shown that microRNA (miRNAs) are involved in molecular pathways responsible for aging and age-related cognitive decline. However, there is a lack of research linked plasma exosome-derived miRNAs changes with cognitive function in older people and aging, which might prove a new insight on the transformation of miRNAs on clinical applications for cognitive decline for older people.

Methods

We applied weighted gene co-expression network analysis to investigated miRNAs within plasma exosomes of older people for a better understanding of the relationship of exosome-derived miRNAs with cognitive decline in elderly adults. We identified network modules of co-expressed miRNAs in the elderly exosomal miRNAs dataset. In each module, we selected vital miRNAs and carried out functional enrichment analyses of their experimentally known target genes and their function.

Results

We found that plasma exosomal miRNAs hsa-mir-376a-3p, miR-10a-5p, miR-125-5p, miR-15a-5p have critical regulatory roles in the development of aging and cognitive dysfunction in the elderly and may serve as biomarkers and putative novel therapeutic targets for aging and cognitive decline.

Introduction

Aging is a complex biological process associated with variable dysfunction and chronic disease susceptibility (Wagner et al., 2016); for example, aging in human and animal is closely associated with changes in cognitive processes. As people age, their memory and the cognitive response rate will gradually decline (Alexander et al., 2012; Febo & Foster, 2016; Nissim et al., 2017; O’Shea et al., 2016; Wagner et al., 2016; Woods, Cohen & Pahor, 2013). Age-related cognitive declines are uneven; meanwhile, many environmental and biological factors including genes, exercise, diet, inflammation, and stress, were thought to be affect the age of onset and cognitive decline (Barrientos et al., 2010; Cai et al., 2014; Craft et al., 2012; Fan, Wheatley & Villeda, 2017; Foster, 2006; Kumar et al., 2013; Speisman et al., 2013).

By 2050, at least 25% of the population will be over 60 years of age in developed countries. Considering that aging is a significant risk factor for many neurodegenerative diseases and cognitive impairment, there is an urgent need to develop a cost-effective and minimally invasive method for identifying individuals with cognitive decline who, thus could benefit from preventive interventions or early treatment (Burkle et al., 2015). A tremendous effort has been made in last decades to validate unbiased blood-based protein for detecting the process of aging, though results have been markedly inconsistent likely due to pre-analytical factors, assay sensitivity, and affinity of the antibodies (Toledo et al., 2011). Furthermore, the limited trafficking of solutes from the brain into the bloodstream through the blood–brain barrier makes that the blood protein levels are challenging to interpret brain changes (Barbu et al., 2009). Therefore, the identification of more accessible accessed, non-protein biological molecules in body fluids was thought to help promote the prevention of aging-related brain diseases and, finally, increasing life expectancy.

Exosomes, membrane-bound vesicles of 40–100 nm in diameter, are presented in biological fluids. They are released from many cell types into the extracellular space. Lipids and proteins are the main components of exosome membranes, where enriched lipid rafts (Gross et al., 2012; Mathivanan, Ji & Simpson, 2010; Schneider & Simons, 2013). Various nucleic acids have recently been identified in exosomes, including mRNA, microRNAs (miRNAs), and other non-coding RNAs (Sato-Kuwabara et al., 2015). During circulation in the body, the RNA carried by exosomes can be taken up by adjacent or distant cells and thus regulate the activity of recipient cells. Exosomes can transport nucleic acids through the blood–brain barrier, enabling exosome carried nucleic acids as a promising candidate for identifying pathological changes in the brain (El Andaloussi et al., 2013; Tominaga et al., 2015; Wood, O’Loughlin & Lakhal, 2011).

MicroRNAs are a class of 17–24 nt small RNAs, which mediate post-transcriptional gene silencing by binding to the 3′-untranslated region or open reading frame region of target mRNAs (Bartel, 2007). The miRNAs involve in many biological activities, including cell proliferation, cell differentiation, cell migration, disease initiation, and disease progression (Liu et al., 2017; Ma, Teruya-Feldstein & Weinberg, 2007; Png et al., 2012; Tay et al., 2008). miRNAs can stably exist in body fluids, including saliva, urine, breast milk, and blood (Arroyo et al., 2011; Gallo et al., 2012; Hu et al., 2010; Lv et al., 2013; Michael et al., 2010; Zhou et al., 2012); they also can be packed into extracellular vesicles such as exosomes (Tabet et al., 2014; Vickers et al., 2015; Arroyo et al., 2011), which help protecting miRNAs from degradation and further guarantee their stability. Given the transportability of vesicles, the role of miRNAs in exosomes is gaining increasing attention (Harvey et al., 2015; Impey et al., 2010; Jovasevic et al., 2015; Rajman et al., 2017; Serafini et al., 2014).

Indeed, exosomes and their carried mRNAs were detected in body fluids, including serum, urine, and saliva in many studies. Notably, studies suggested that circulating levels of miRNAs in plasma (Kumar et al., 2013) or exosomes (Cheng et al., 2015; Lugli et al., 2015) were capable of identifying Alzheimer’s disease (AD). Several blood-based miRNAs have been reported to be differentially expressed in AD (Cosin-Tomas et al., 2017; Dong et al., 2015; Galimberti et al., 2014; Geekiyanage et al., 2012; Kiko et al., 2014; Kumar et al., 2013; Leidinger et al., 2013; Tan et al., 2014). Alzheimer’s syndrome is a neurodegenerative disease whose typical manifestation is a decrease in cognitive function. Studies have shown that miRNA-146a is associated with high-risk Alzheimer’s syndrome and cognitive decline (Cui et al., 2014) and that hippocampal miRNA-132 can receive stress-induced cognitive impairment (Shaltiel et al., 2013). In this study, we investigated whether plasma exosome derived miRNAs are associated with changes in age-related cognitive functioning dysfunction. Our study may contribute to understanding the mechanisms beneath the age-related cognitive decline and selecting mRNAs as a biomarker for identifying this kind of disease.

Materials and Methods

Data collection and preprocessing

The miRNA-seq based expression dataset GSE97644 was downloaded from the NCBI Gene Expression Omnibus (Rani et al., 2017). Data analyses were performed using the R programing language (v3.5.0), Bioconductor packages (v3.7), and R functions in the weighted gene co-expression network analysis (WGCNA) package (Langfelder & Horvath, 2008). The CPM normalized expression data were quantile normalized for subsequent data analysis.

Weighted gene co-expression networks and their modules

To explore the relationship between changes in miRNA expression with age and The Montreal Cognitive Assessment (MoCA) scores, we used WGCNA. WGCNA is a useful method to reveal the relationship between linking clustered genes and phenotypic traits. In vivo, miRNAs act in the same way as mRNA and are also consistent with the scale-free topology. Therefore, we use the WGCNA method to analyze the relationship between link clustered miRNAs and phenotypic traits (Giulietti et al., 2017; Yepes et al., 2016). The MoCA is a 10-min cognitive screening tool to assist first-line physicians in the detection of mild cognitive impairment (MCI), a clinical state that often progresses to dementia. The MoCA is a brief cognitive screening tool with high sensitivity and specificity for detecting MCI (Nasreddine et al., 2004).

Select the optimal soft threshold

The research shows that the co-expression network conforms to the scale-free network (Langfelder & Horvath, 2008), that is, the logarithm log(k) of the node with the connection degree k is negatively correlated with the logarithm log(p(k)) of the probability of the node, and the correlation coefficient is higher than 0.8. R software package WGCNA was used to build a weighted co-expression network. To ensure that the network is a scale-free network, we choose a soft threshold of β = 10.

Construct co-expression modules

Convert the expression matrix into an adjacency matrix, and then convert the adjacency matrix into a topological matrix. Based on TOM, the average-linkage hierarchical clustering method was used to cluster the miRNAs according to the criteria of the mixed dynamic cut tree and set each miRNA. The network module has a minimum base number of 20. After using the dynamic shear method to determine the miRNA module, we calculate the eigengenes of each module in turn, then cluster the modules to merge the closer modules into new modules, and set height = 0.2.

Phenotypic correlation analysis of modules

According to the feature vector of each module, we calculate the correlation between these modules and each phenotype separately. The module-trait relationship was estimated using the correlation between the module eigengene and the phenotype (gender, age, and MoCA score). For each expression profile, miRNA significance was calculated as the absolute value of the correlation between expression profile and each trait; module membership (MM) was defined as the correlation of expression profile and each module eigengene (Shi et al., 2015).

Analysis of miRNAs and phenotype correlation of modules

Based on the expression levels of miRNAs of each sample, we calculated the correlation between the miRNAs in these modules and each phenotype to measure the degree of association between miRNAs and phenotypes. The larger the value, the more biologically meaningful. The correlation equal to 0 indicates that the miRNA is not related to the phenotype.

Correlation between integration modules and phenotypes and miRNA and phenotypic correlation in modules

Our integration module and phenotypic correlation results and miRNA- and phenotypic correlation results in the module further analyze the correlation between the two, and then speculate the relationship between miRNAs and phenotype, the higher the correlation, the phenotype more relevant to the module.

Module-related miRNA correlation analysis

According to the eigenvectors of each module, we calculated the correlation between the expression of these modules and the miRNAs in their modules. Among them, the miRNAs with a correlation higher than 0.85 in each module (Hub miRNAs) were selected for subsequent target gene prediction and functional annotation.

Hub miRNAs and their functional annotations

To explain the biological functions of the miRNAs in these modules, we performed functional enrichment analysis of the targets of these miRNAs. We imported the selected miRNAs into the miRNet web tool (Fan & Xia, 2018) to enrich the corresponding miRNA target genes in biological pathways, processes, and molecular functions. In particular, this tool can perform functional enrichment of Kyoto Encyclopedia of Genes and Genomes (KEGG) and REACTOME pathways based on experimentally validated miRNA targets. Therefore, the signal path enriched by this tool has higher reliability.

Results

Weighted gene co-expression network analysis

Firstly, we obtained the expression profile and the phenotypic database, which contained 97 samples, 2,588 miRNAs, and three phenotypes. We removed the miRNAs with NA expressing records according to the expression profile, calculated the variance of each miRNA in each sample, and selected the miRNAs with a standard deviation higher than 0.5. All the samples further clustered, as shown in Fig. 1. We removed the unreasonable samples according to the cluster distance, as shown in the red line in the figure. Finally, we got a new data expression spectrum, containing a total of 91 samples, 302 miRNAs. We chose a soft threshold of β = 10 in order to ensure building the scale-free characteristics of the network (Fig. 2).

Figure 1 Sample clustering to detect outliers.

In order to make the dataset more reasonable, The hierarchical clustering method was used to cluster the expression profile of the samples and calculated the distance between the samples. X1593VS, X1830DX, X1530LXD, X1743CB, X1663FXC, X1701AA, these six samples were removed according to the cluster distance, as shown in the red line in the figure.

Figure 2 Analysis of network topology for various soft-thresholding powers bd c9.

(A) The scale-free fit index (y-axis) as a function of the soft-thresholding power (x-axis). (B) The mean connectivity (degree, y-axis) as a function of the soft-thresholding power (x-axis).

After Computational co-expression modules, a total of five modules are shown in Fig. 3, where the gray module is a collection of miRNAs that could not be aggregated into other modules.

Figure 3 Clustering dendrogram of miRNAs.

MiRNAs with higher topology similarity were clustered into different modules and assigned different colors. The expression profiles of miRNAs in each module had similar structural features in the network topology. Four co-expression modules were constructed and shown in different colors. These modules were ranged from large to small by the number of genes they included. The number of miRNAs in the four modules were listed in Table 1.

The miRNA statistics in each module are shown in Table 1, and it can be seen that these miRNAs are assigned to five modules. The blue, brown, turquoise, yellow, and gray module contains 40, 33, 44, 27, and 158 miRNAs respectively.

Table 1 The co-expression modules and hub miRNAs in each module (Correlation coefficient > 0.85).

Module	Blue module	Brown module	Turquoise module	Yellow module	
Total miRNAs	40	33	158	27	
hub miRNAs	hsa-miR-376c-3p	hsa-miR-125b-5p	hsa-let-7i-5p	hsa-miR-1307-3p	
	hsa-miR-376a-3p	hsa-miR-10b-5p	hsa-miR-340-5p	hsa-miR-378a-3p	
	hsa-miR-369-5p	hsa-miR-99a-5p	hsa-miR-15a-5p	hsa-miR-320b	
	hsa-miR-654-3p	hsa-miR-100-5p	hsa-miR-374b-5p	hsa-miR-671-5p	
	hsa-miR-487b-3p	hsa-miR-483-3p	hsa-let-7g-5p	hsa-miR-23a-5p	
	hsa-miR-495-3p	hsa-miR-214-3p	hsa-miR-454-3p	hsa-miR-1908-5p	
	hsa-miR-496	hsa-miR-342-3p	hsa-miR-30b-5p		
	hsa-miR-329-3p	hsa-miR-10a-5p	hsa-miR-590-3p		
	hsa-miR-136-3p	hsa-miR-125a-5p	hsa-let-7f-5p		
	hsa-miR-431-5p				

For the Phenotypic correlation analysis of modules, we analyze the relationship between the feature vectors of each module with the gender, age, and MoCA score.

The correlation between the phenotype and each module showed in Fig. 4. The blue module and the turquoise module have a positive correlation with age. Thorough Analysis of miRNA and phenotype correlation of modules, we separately calculated the distribution of GS of each phenotype in each module. Figures 5A–5C illustrated the overall correlation between the phenotype and the miRNAs in each module.

Figure 4 Module-trait associations.

Each row corresponds to a module eigengene, column to a trait. Each cell contains the corresponding correlation and p-value. The table is color-coded by correlation according to the color legend. The results showed that the blue module was more correlated with age (p = 0.06), and each module was less correlated with gender.

Figure 5 MiRNAs and phenotype correlation of modules.

(A–C) Correlation between integration modules and phenotypes and miRNA and phenotypic correlation in modules. The average miRNA significance in the blue module, the turquoise module, and the yellow module was highly correlated with age; the average miRNA significance in the turquoise module and the brown module was highly correlated with the MOCA score. (D–O) Analysis of miRNAs and phenotype correlation of modules. From the analysis results in the figure, it was known that age has a high correlation with the miRNA in the blue module (p < 0.001) and the brown module (p < 0.05). The MOCA score was highly correlated with the miRNA in the blue module (p < 0.01).

By integrating the analytical data of correlation between integration modules and phenotypes, as well as miRNA and phenotypic correlation in modules, we further analyzed the relationship between MM and miRNA significance for each module in each phenotype. The corresponding analysis results for each phenotype are as shown in Figs. 5D–5O. In the gender feature, the correlation between MM and miRNA significance of each module is very low, proving there is no significant relationship between the four modules and gender. In the age feature, we find that the blue module is involved in the regulation of this feature (cor = 0.57, p = 0.00012). In the MoCA score feature, we found that the brown module was involved in the adjustment of this feature (cor = 0.31, p = 0.079). Through Module-related miRNA correlation analysis, we selected miRNAs with module correlation greater than 0.85 in each module from each module. These miRNAs were used for target gene prediction and functional enrichment analysis.

Hub miRNAs and their functional annotations

To clarify the link between each module and the trait, we selected the miRNAs with the highest MM scores in each module for functional enrichment. We selected 10 miRNAs from the blue module, nine miRNAs in the brown module, nine miRNAs in the turquoise module, and six miRNAs in the yellow module (Table 1). We performed functional enrichment analysis and were limited to experimentally confirmed miRNA target genes. The enriched KEGG and REACTOME pathways were identified by analyzing the hub miRNAs of each module using the miRNet network tool. The hub miRNAs in the blue module are mainly enriched in aging-related signaling pathways such as Aging, Cellular responses to stress, and Oncogene induced Senescence (Figs. 6A and 6B). The hub miRNAs of the Brown module are mainly enriched in Pathways in cancer, Gene Expression, disease, aging, protein complex disassembly (Figs. 6C and 6D). The hub miRNAs in the Turquoise module are mainly enriched in cell circle, TGF-beta signaling pathway, apoptosis, immune system (Figs. 6E and 6F). The hub-miRNAs in the Yellow module are mainly enriched in miRNA synthesis-regulated, senescence-related pathways (Figs. 6G and 6H).

Figure 6 Functional annotation of hub miRNAs in each module ((A and B) blue module; (C and D) brown module; (E and F) turquoise module; (G and H) yellow module): enriched KEGG pathways (A, C, E, G) and REACTOME pathways (B, D, F, H).

The y-axis represents the metabolic pathway in which the target genes of the hub miRNAs are involved, and the x-axis represents the ratio of Hits value of the mRNA enriched in each metabolic pathway to all mRNAs in this metabolic pathway. The size of the circle represents the Hits value of the target gene of the microRNA, and the color of the circle represents the significance of the signal pathway in which the target gene is involved (–log10(p)), and the redder color indicates a significantly higher value. The hub miRNAs in each module were shown in Table 1.

We used miRNet web tools to build miRNA-target interaction networks. The hub miRNAs (module and miRNA p-value > 0.85) in each module were used as the input node to calculate the node degree and betweenness, both of which indicators for node centrality. We found that hsa-mir-329-3p in the blue (Fig. 7A) has the highest 538 Degree, and the target genes in this module are AGO2, BTF3L4, WDR17, BMI1, UHMK1, TIPRL, AKT1, ARPP19, IGF1R with three degrees. In the brown module (Fig. 7B), hsa-mir-10a-5p, has-mir-125b-5p have 463,432 degrees, respectively, among the target genes in this module, AKT1, ANKRD33B, PLA2G4F, TMEM101, IGF1R, EIF1AD, GSS, SREBF1, TP53, EXTL3, RUNDC3B, MKNK2, CCNG1, CSNK2A1, XIAP, NCOR2, PAFAH1B1, SNX4 have at least four degrees. In the turquoise module (Fig. 7C), hsa-mir-15a-5p has the highest 717 degrees, of which KEMEN1, CCND1, KLHL15, PPP1R15B, ZNF264, ZNF460, YOD1, NOM1, WASL, TNFSF9, NCOA3 have more than five degrees. In the yellow module (Fig. 7D), hsa-mir-1307-3p has the highest 240 degrees, of which PEX26, ZNF385A has more than three degrees. From these results, we can think that miRNAs in exosomes can play a regulatory role in inter-tissue signaling. In particular, miR-329-3p, miR-10a-5p, miR-125b-5p, miR-15a-5p play an important role in the regulation of exocytosis transmission, and these functions are closely related to the decline of aging and cognitive functions (Bottoni et al., 2005; Tan et al., 2014).

Figure 7 Interaction network of hub miRNAs.

Interaction network of hub miRNAs in blue (A) and brown (B) and turquoise (C) and yellow (D) with their experimentally validated target genes using the miRNet tool. This network graphically high-lights which miRNAs exhibit a large number of interactions.

Discussion

In this study, data came from 97 elderly patients. Four co-expression modules were analyzed by WGCNA analysis. WGCNA can be used to explore the relationship between each module and clinical traits. WGCNA analysis focuses on the Analysis of the link between co-expression modules and clinical traits. Therefore, compared to other analytical methods, the analysis results of WGCNA have higher credibility and biological significance (Chou et al., 2014). Genes in the same block are considered to be functionally related. Therefore, miRNAs in modules that are highly correlated with biological traits can be considered as biomarkers for clinical testing and treatment. We analyzed four co-expression modules. In these modules, we found that the blue module has a specific correlation with age. Although the expression profiles of these miRNAs are relatively low in covariance correlation with Age and MoCA score, exosomes can be used as an indicator of brain health as a membrane molecule that can cross the blood–brain barrier (Andras & Toborek, 2016). At the same time, exosomes in the blood can transport functional macromolecules such as proteins, mRNAs, and miRNAs to the brain through the blood–brain barrier, thus affecting the function of the brain. In particular, it is considered that miRNAs can be enriched in exosomes so that the concentration of miRNA in exosomes is much higher than the concentration of miRNA in normal blood. Therefore, exosomes achieve regulation of the brain, which may be an important means of regulating the brain’s function through humoral regulation (Bastos et al., 2017). At the same time, recent research has also shown that the hypothalamic-mediated aging process is accompanied by dysregulation of nutrient sensing, altered intercellular communication, stem cell exhaustion, loss of proteostasis, and epigenetic alterations. This suggests that the process of aging is strictly regulated by the central nervous system of the brain. From these perspectives, exosomes in the blood may be important hubs for the regulation of aging and cognitive decline in the blood circulatory system and brain, and miRNAs in exosomes may play an important role. In order to further explore the relationship between the co-expression module of miRNAs in exosomes and the age and MoCA scores, we used KEGG and Reactome to perform the enrichment analysis for the hub miRNAs in each module. From the results, the antigen of blue module hub miRNAs, mir-376a-3p, is mainly involved in tumor-associated metabolic pathways (Gao et al., 2010; Tu et al., 2016) and muscular dystrophy. However, Thalyana Smith-Vikos et al. found that the plasma content of hsa-mir-376a-3p in long-lived people was significantly different from that in short-lived people, indicating that mir-376a-3p is strongly associated with age. Mir-369-5p is involved in the regulation of senescence of mesenchymal stem cells (Wagner et al., 2008) and participates in Duchenne muscular dystrophy (Bottoni et al., 2005), Nemaline myopathy (NM) and other muscular dystrophy associated disease (Xia et al., 2008). Muscle atrophy is a typical feature of aging. Yao et al. (2019) found that the up-regulation of miR-496 can reduce cerebral ischemia-reperfusion injury. SP Ross at al found that miRNA-431 can prevent synaptic loss in amyloid-β-induced cell culture models of AD by silencing the expression of Kremen1 protein (Ross et al., 2018). Among the brown module, the target genes of hub miRNAs are mainly involved in cancer-related pathways. Mir-125b-5p is involved in Alzheimer Disease (Sethi & Lukiw, 2009), Cerebellar neurodegeneration disease metabolic pathway, and it is also related to the occurrence of vascular diseases and heart failure (Dai et al., 2009), and there is a strong correlation between the two diseases with the life expectancy of the elderly. Mir-342-3p is involved in Neurodegeneration and Prion diseases (Sayed et al., 2007). Among the turquoise modules, hub miRNAs are mainly involved in cancer-related signaling pathways. The let7 family of miRNAs (let-7i-5p, let-7g-5p, let-7f-5p) are thought to be highly correlated with the development of cancer and are important mechanisms for the body to suppress cancer (Pobezinsky & Wells, 2018). Among the yellow module, the target genes of hub miRNAs are also mainly enriched in cancer-related pathways. We obtained five miRNAs from these modules, miR-376a, miR-125a, miR-10a, miR-15a. Among them, miR-37a is up-regulated in T lymphocytes of patients with multiple sclerosis (Lindberg et al., 2010), miR125a is up-regulated in PBMC of patients with multiple sclerosis (Yang et al., 2014), miR-10a and miR-125 are down-regulated in cerebrospinal fluid of patients with Alzheimer’s syndrome (Bekris et al., 2013), miR-15a Up-regulation of activated lesions in brain lesions in patients with multiple sclerosis is an age-related disease (Junker et al., 2009). These findings support the role of our selected hub miRNAs in Aging and age-related diseases. The hub miRNAs in these modules are associated with aging and neurodegenerative diseases. However, their expression profiles are relatively low in correlation coefficients between age and MoCA scores, making it difficult to find definitive connections. This also shows that the aging and cognitive impairment of the body are the results of many factors.

Furthermore, we identified enriched KEGG and REACTOME pathways through functional enrichment analysis of hub miRNA target genes. In this way, we found the relationship between the four module-related signaling pathways and the aging and MoCA scores. Interesting, we found that these modules are all related to cancer-related signaling pathways. Cancer is an important factor affecting the longevity of the elderly (Leshno et al., 2016; Paskett et al., 2018).

We also identified the highest number of genes targeted by hub miRNAs in each module (Fig. 7). In the blue module, the ARPP19 was found to be highly targeted. The 19-kD cAMP-regulated phosphoprotein (ARPP19) plays a role in regulating mitosis by inhibiting protein phosphatase-2A (PP2A) (Gharbi-Ayachi et al., 2010). Decreased levels of ARPP-19 and PKA in brains of Down syndrome and AD (Kim et al., 2001). Interesting, ARPP19 expression in brain tissue is significantly higher than other tissues (Fagerberg et al., 2014). Within the brown module, we found that the AKT1, RUN domain containing 3B (RUNDC3B), cyclin G1 (CCNG1), platelet-activating factor acetylhydrolase 1b regulatory subunit 1 (PAFAH1B1), tumor protein p53 (TP53), sorting nexin 4 (SNX4) The expression levels of RUNDC3B and PAFAH1B1 in brain tissue are significantly higher than in other tissues (Fagerberg et al., 2014), where PAFAH1B1 gene duplication is associated with developmental, behavioral and brain abnormalities (Curry et al., 2013). SNX4 gene encodes a member of the sorting nexin family. Study shows that SNX4-mediated regulation of the steady-state levels and trafficking of BACE1, as well as the subsequent increase in BACE1-mediated cleavage, may be relevant to AD progression (Kim et al., 2017). These genes are primarily involved in the regulation of cancer signaling (Cruz-Garcia et al., 2018; Ko & Kim, 2019; Ma, Lu & Gu, 2019). Within the turquoise module, the protein phosphatase 1 regulatory subunit 15B (PPP1R15B), TNF superfamily member 9 (TNFSF9), nuclear receptor coactivator3 (NCOA3) were found to be highly targeted. The TPSF9 is a cytokine that belongs to the tumor necrosis factor (TNF) ligand family. This transmembrane cytokine is a bidirectional signal transducer, acts as a ligand for TNFRSF9/4-1BB, which is a costimulatory receptor molecule in T lymphocytes (Tsuda et al., 2017). The NCOA3 is a nuclear receptor coactivator that interacts with nuclear hormone receptors to enhance their transcriptional activator functions. Data suggest that over-stimulating the steroid receptor coactivators SRC-1, SRC-2, and SRC-3 oncogenic program can be an effective strategy to kill cancer cells (Wang et al., 2015). Within the yellow module, we found that the zinc finger protein 385A (ZNF385A) was highly targeted. Brain gene expression of ZNF385A and DNA methylation dysregulations are implicated in the Alteration of brain tissue properties associated with late-life cognitive decline above and beyond the influence of common neuropathologic conditions (Yu et al., 2017). We analyzed the experimentally validated interactions by miRNA-target network and eliminated the network. The node with degree = 1 intensifies the interaction of the internal interactions in each module and, of course, also loses much information about the possible interactions between miRNA-mRNA.

Conclusions

In summary, we found four linking clusters miRNA expression profiles from the plasma exosome of human peripheral blood and analyzed target gene prediction and functional enrichment for hub miRNAs in these four clusters. The relationship between modules and age and cognitive function. Some of these hub miRNAs have been experimentally validated for their relationship with age and cognitive function, and some new candidate miRNAs that may be exploited as biomarkers or therapeutic targets of aging or cognitive decline. Although there is overlap between known candidate biomarkers and the ones that we identified here, more research needs to be used to verify the performance of these possible biomarkers.

Supplemental Information

Supplemental Information 1 The main principle of WGCNA algorithms.

Click here for additional data file.

Supplemental Information 2 Hub miRNA target gene map (minus branches).

We subtracted the target gene with the node equal to 1 from the network map and obtained the cut-down network map. Each Figure (A–D) corresponds to the subgraph in Fig. 7.

Click here for additional data file.

Supplemental Information 3 The KEGG, REACTOME, GO Enrichment Analysis Results of the hub miRNAs in each module.

Click here for additional data file.

Supplemental Information 4 Overall schematic of the article.

Click here for additional data file.

Supplemental Information 5 The miRNA expression matrix for each sample.

Click here for additional data file.

Supplemental Information 6 The sex, MoCA score and age information of the samples.

Click here for additional data file.

We thank Doulathunnisa Jaffar Ali for help in honing the manuscript.

Additional Information and Declarations

Competing Interests

Author Contributions

Data Availability

The authors declare that they have no competing interests.

Zheng Ye analyzed the data, conceived and designed the experiments, performed the experiments, prepared figures and/or tables, authored or reviewed drafts of the paper, and approved the final draft.

Bo Sun analyzed the data, prepared figures and/or tables, authored or reviewed drafts of the paper, and approved the final draft.

Xue Mi performed the experiments, authored or reviewed drafts of the paper, and approved the final draft.

Zhongdang Xiao conceived and designed the experiments, authored or reviewed drafts of the paper, and approved the final draft.

The following information was supplied regarding data availability:

The raw measurements are available in the Supplemental Files.

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
