# Peer review of "Gene co-expression network for analysis of plasma exosomal miRNAs in the elderly as markers of aging and cognitive decline"

_PeerJ, doi:10.7717/peerj.8318_

## Round 0.1 · original submission · Major Revisions

Three specialists in the field recommended revisions in the manuscript. One of the reviewers requested some revision on English usage. The authors carried out data analysis using the R programming language (v3.5.0), Bioconductor packages (v3.7) and R functions in the WGCNA package. In my view, data analysis is adequate. One minor point: Throughout the manuscript, the authors used the red font in a few sentences. Please use the same type of font color.

Reviewer 1 ·

Basic reporting

Authors have applied weighted gene co-expression network analysis (WGCNA) to investigated miRNAs within plasma exosomes of elder people. Using hub node and pathway enrichment analysis they further narrow down four microRNAs (hsa-mir-376a-3p, has- miR-10a-5p, has-miR-125-5p, has-miR-15a-5p) that have regulatory roles in the development of ageing and cognitive dysfunction. Considering ageing is a complex phenomenon, the prediction of biomarker only focusing on exosomal miRNAs is overstated.

Experimental design

The analysis was done using the datasets available on the website. The tools they have used for miRNAs target prediction are also web-based tool. Details parameter for analysis is lacking in each step. Statistical analysis details are also missing.

Validity of the findings

The results are solely bioinformatics based, require experimental validation in a large number of patients before making use as a biomarker for ageing.

Additional comments

Authors have applied weighted gene co-expression network analysis (WGCNA) to investigated miRNAs within plasma exosomes of elder people. Using hub node and pathway enrichment analysis they further narrow down four microRNAs (hsa-mir-376a-3p, has- miR-10a-5p, has-miR-125-5p, has-miR-15a-5p) that have regulatory roles in the development of ageing and cognitive dysfunction. Overall, this is a good effort to identify biomarkers using bioinformatics tools. However, the manuscript lacks in-depth details of the methods. The inclusion and exclusion criterion choosing the dataset is missing. A brief of Pipeline describing the steps and criteria followed will give a more clear idea about data analysis. More importantly, considering the complex nature of the ageing, how did the authors deal with the false positivity? What cut off values they have used choosing miRNAs from different modules.

Few additional issues:
1. Line 92 : Which correlation calculation method you have applied?

2. Line 195 : …… data came from 79 elderly patients. In Line 137 …. 97 samples. Which one is true?

3. Line 245 : ….. identified the highest number of genes targeted by hub miRNAs
How did you perform this?

Reviewer 2 ·

Basic reporting

Paper refers to plasma exosome -derived miRNAs changes and its relation with cognitive function in older people. In general, the paper is well-written, however, and english proofreader will improve importantly the reading. There are some miscues in the references since some of them have DOI and others don´t please try to make it more homogeneous. Some figures have a title and others do not, please correct, additionally add more explanation to figures ( #1, #3, #4, #5 ) so they could be more easy to understand. In cases of Figure 6 and 7, please put the most representative images and add the others to supplementary figures, otherwise, they are quite difficult to follow. In the case of Table 1, perhaps it could be added somehow to Table 2 since it´s quite small and has very few information. In the introduction, it is not stated why the cognitive decline is related to miRNA´s profiles? please add more information on the subject.

Experimental design

Please, add more information on the WGCNA package. It´s not clear what was the experimental procedure performed in the papers, since authors state that they investigated miRNAs within plasma exosomes of older people, but it is not stated in methodology the blood extraction or the exosome isolation, otherwise please clarify.

Validity of the findings

In general, it is not clear the experimental analysis of exosomes and miRNAs, since this study is a bioinformatic analysis, authors should specify such relation. Discussion is quite big please try to diminish information and discuss the differences of your findings with other papers and clinical importance of such.

Additional comments

In general, this paper seems to be a complete bioinformatic analysis, however, experimental relation between exosomes and miRNA´s is not clear. Additionally, there are miscues that difficult reading, so perhaps proofreader could be useful. Figures need more information so they could be easy to understand. Reference needs to be more homogeneous.

Reviewer 3 ·

Basic reporting

In general it is well raised, even with the suggested corrections.

But I have some suggestions:

In the introduction the relationship between miRNA and aging is poorly established.

Line 195: It says:

In this study, data came from 79 elderly patients

Was it mentioned before?

Experimental design

The methodology part is well raised

Validity of the findings

the findings are conclusive, it would have been important that trials are designed to validate the miRNA found as prevalent: hsa-mir-376a-3p, miR-10a-5p, miR-125-5p, miR-15a-5p

Additional comments

They are mentioned above

---

## Round 0.2 · accepted · Accept

The authors carried out substantial modifications in the revised version of the manuscript. The overall quality of this submission improved a great deal and can be accepted for publication as it is.

Reviewer 3 ·

Basic reporting

The paper addresses a very current theme.

It is very well written, with references that provide what they point out

only comments that are less than:

The introduction has different font sizes.

There is no consistency of how they write miRNA or even how they cite references

Experimental design

they used the right tools

Validity of the findings

each result is analyzed very well also serves to build the following result